# Precision Oncology Targets in Biliary Tract Cancer

**DOI:** 10.3390/cancers15072105

**Published:** 2023-03-31

**Authors:** Nicole Farha, Danai Dima, Fauzia Ullah, Suneel Kamath

**Affiliations:** 1Cleveland Clinic Foundation, Cleveland, OH 44195, USA; 2Department of Hematology/Oncology, Cleveland Clinic Taussig Cancer Institute, Cleveland, OH 44195, USA; dimad@ccf.org (D.D.);

**Keywords:** cholangiocarcinoma, biliary cancer, targeted therapy, second line, treatment resistant, advanced stage, monotherapy, combination therapy

## Abstract

**Simple Summary:**

Biliary tract cancer is rare in the US but remains a highly lethal cancer. While the standard of care in the first-line, metastatic setting is well-established, no second-line regimen has been clearly defined as the gold standard. Many novel biomarker-targeted therapies have emerged and shown promise, even in treatment-refractory patients. Here, we seek to provide updates on the landscape of targeted therapies available for the treatment of advanced biliary tract cancer.

**Abstract:**

Targeted therapies in biliary tract cancer (BTC) are emerging as options for patients not who do not respond to first-line treatment. Agents acting on tumor-specific oncogenes in BTC may target fibroblast growth factor receptor 2 (FGFR2), isocitrate dehydrogenase (IDH), B-raf kinase (BRAF), and human epidermal growth factor receptor 2 (HER-2). Additionally, given the heterogeneous genetic landscape of advanced BTCs, many harbor genetic aberrations that are common among solid tumors, including RET fusions, tropomyosin receptor kinase (TRK) fusions, and high tumor mutational burden (TMB). This review aims to provide updates on the evolving array of therapeutics available, and to summarize promising works on the horizon.

## 1. Introduction

Advanced biliary tract cancer (BTC) represents a relatively rare, heterogeneous group of cancers, consisting of cholangiocarcinoma (CCA), including intrahepatic cholangiocarcinoma (iCCA) and extrahepatic cholangiocarcinoma (eCCA), and gallbladder carcinoma (GBC) (Figure 1). Globally, iCCA accounts for 10–20% of BTC, eCCA represents 20–30%, and GBC makes up the bulk of BTC at 50–60% [1,2]. Analyses of SEER data estimate that 4 in 100,000 people will be diagnosed with CCA each year in the US [2], and the incidence is increasing [1]. This contrasts with certain regions of Korea, China, and Thailand, where the incidence is higher [3]. Nevertheless, survival is grim, with 5-year survival rates reaching as low as 2.5% in distant-stage cancers [4]. When separated by primary site, studies have found that the median overall survival is universally dismal: 6 months for iCCA, 9 months for eCCA, and 2 months when the primary site was unidentifiable [3]. GBC is more common, and more likely to be discovered at a localized stage, conferring better rates of survival [4]. However, once spread, it behaves very similarly to CCA; thus treatment guidelines for the advanced stage often group them together.

Given its asymptomatic nature, BTC is often identified after progression, complicating treatment [3]. Therapeutic modalities are limited to surgery, radiation, chemotherapy, and targeted agents, however many individuals will not be candidates for these given their poor state of health following a late-stage diagnosis, and treatment failure is common. Surgery remains the only curative option, but despite surgery proving to prolong survival [5], tumor resectability is highly variable and depends on a myriad of factors including stage, ductal involvement, presence of portal vein compromise, and degree of hepatic atrophy [6]. Resectability rates are particularly low for CCA, estimated around 30% [7], and even after resection, studies have found recurrence rates of over 75%, often within the first two years [8,9]. For example, in the BILCAP trial, the median recurrence-free survival with adjuvant capecitabine was just 25.9 months [10]. Given this, systemic therapy plays an important role.

First-line systemic agents include gemcitabine and cisplatin, based on the ABC-02 study [11]. Recently, the addition of durvalumab was approved based on the TOPAZ-1 trial, as the triplet demonstrated a significant improvement in overall survival and response rates compared to gemcitabine and cisplatin alone [12,13]. The National Comprehensive Cancer Network (NCCN) Clinical Practice Guidelines in Oncology encourage some genome-based therapies for the first line in select cases [14], underscoring the importance of early molecular profiling. However, despite these promising therapies, advanced BTC remains a treatment-refractory disease; more than 50% of participants in the TOPAZ-1 trial experienced progression within the first 7.5 months of treatment [12], emphasizing the importance of strong second-line therapies.

Targeted therapy is primarily employed in cases of chemo-refractory BTC. Several well-studied oncogenic pathways are implicated in BTC, especially in iCCAs, which harbor more actionable mutations when compared to eCCAs and GBCs, which are known to express more undruggable alterations [15]. Here we aim to outline the genomic targets and their therapeutic complements approved for the treatment of advanced, metastatic, and chemo-resistant BTC (Table 1), as well as those still under investigation.

## 2. Tumor-Specific Treatments

### 2.1. Fibroblast Growth Factor Receptor (FGFR) Inhibitors

Fibroblast growth factors (FGFs) produce their biological actions by signaling through FGFRs, which mediate cellular proliferation, differentiation, and survival, as well as angiogenesis [16]. By constitutively activating the FGFR receptor through gene amplification, mutation, or rearrangement, cancer cells can proliferate without limitation and evade the body’s immune safeguards. 

FGFR1 alterations are the most common among solid tumors, occurring in 49% of all FGFR-mutant tumors [16]. In BTC specifically, FGFR2 mutations are the most relevant FGFR alteration. They occur in 10–15% of iCCA, and are very rare in eCCA [17], making them the second most common targetable mutation behind IDH. Despite this, the efficacy of FGFR2-directed therapy makes FGFR2 inhibition likely the most clinically promising targeted therapy in the disease.

Pemigatinib and futibatinib are the two currently available inhibitors of FGFR2 (FGFR2Is), recommended for patients with BTC harboring FGFR2 fusions or rearrangements in the second line setting and beyond [14]. Pemigatinib is a selective oral reversible inhibitor of FGFR1-3, approved after the phase 2 FIGHT-202 trial demonstrated its efficacy, with a 35.5% overall response rate (ORR) (95% CI 26.5–45.4) including three complete responses (CRs) in advanced BTCs that had failed prior therapy [18]. Median overall survival (mOS) was 21 months, although not yet mature at the time of study completion (95% CI 14.8-not estimable). Significant adverse events primarily included hyperphosphatemia, hypophosphatemia, ocular toxicity, hand–foot syndrome, and onycholysis, with the remainder being uncommon. The phase 3 randomized FIGHT-302 clinical trial comparing pemigatinib to gemcitabine and cisplatin as first-line treatments in advanced BTC is currently underway and highly anticipated (NCT03656536). 

Futibatinib is another potent oral agent that stands out as a covalent irreversible pan-inhibitor of FGFR1-4. Futibatinib was granted accelerated FDA approval [19] based on data from the phase 2 FOENIX-CCA2 trial showing an ORR of 41.7% and a 12-month survival rate of 73% [20]. Futibatinib has a very similar toxicity profile compared to pemigatinib, with the most common toxicities including hyperphosphatemia, ocular toxicity, hand–foot syndrome, nail toxicity and stomatitis [18]. Given its promising efficacy and safety profile, the FOENIX-CCA3 trial (NCT04093362) is ongoing, investigating futibatinib’s potential for first–line therapy when compared to gemcitabine and cisplatin. 

A third pan-FGFR inhibitor, infigratinib, was voluntarily withdrawn from the market by its manufacturer [21]. This was possibly due to its inferior activity (ORR: 23.1%) compared to pemigatinib and futibatinib [22], although no comparative trials have been conducted. 

RLY-4008 is the first highly selective inhibitor of FGFR2 designed to limit off-target toxicity and to overcome polyclonal FGFR2 resistance mechanisms [23]. Preliminary data from the phase 1/2 ReFocus trial (NCT04526106) presented at the ESMO Congress 2022 demonstrated potent efficacy, with an ORR of 88% (95% CI 63.6–98.5) in the first 17 patients with FGFRi-naïve BTC who received the recommended phase 2 dosing [23]. The drug was well tolerated overall, with only low-grade adverse events such as stomatitis, PPE, dry mouth and ocular toxicity. Further expansion of this cohort, along with enrollment in additional cohorts for FGFR-inhibitor-exposed BTCs and all solid tumors are ongoing. 

### 2.2. Isocitrate Dehydrogenase (IDH) Inhibitors

Mutations in IDH genes are thought to occur in approximately 20% of cases of iCCA, making them the most common targetable mutation in BTC [24]. Although the exact oncogenic mechanism is unclear, mutant IDH has been associated with the accumulation of oncogenic metabolites and possible immunosuppressants [2]. 

IDH1 and IDH2 are the genes encoding the IDH enzyme proteins, and their mutations have been well-studied in AML and glioblastoma [25]. IDH is an essential enzyme for cellular respiration in the TCA cycle, catalyzing the decarboxylation of isocitrate to alpha-ketoglutarate and CO_2_ [2]. Mutations are typically gain-of-function, with the most common among them being the IDH1 R132C mutation; they lead to disruption of the normal catalytic activity, resulting in increased conversion of alpha-ketoglutarate to D-2-hydroxyglutarate, which is a known oncometabolite, promoting tumor proliferation and metastasis through several pathways [2].

The NCCN currently recommends ivosidenib, an oral inhibitor of IDH1, in the second-line treatment of advanced, unresectable, and metastatic BTCs of all sites that harbor this mutation [14]. This guideline came after FDA approval of the drug based on the phase 3, randomized, placebo-controlled CLARIDHY trial (NCT02989857) [26]. The initial results were disappointing, with the intention-to-treat analysis demonstrating significantly improved progression-free survival (PFS) in the treatment arm, but no significant difference in overall survival (OS), and an underwhelming ORR of 2.4%, representing three partial responses [27]. However, the results adjusting for crossover into the treatment arm were able to demonstrate modest improvement in OS (HR 0.49, 95% CI 0.34–0.70), as well as in PFS (HR 0.37, 95% CI 0.25–0.54) [28]. The most common treatment-related adverse event (TRAE) was ascites, seen in 11 (9%) treated patients and 4 (7%) in the placebo arm, while their quality of life appeared unchanged. 

Ongoing phase 1 trials are investigating ivosidenib as a part of first-line combination therapy with gem-cis (NCT04088188), slated for completion in 2025. Other IDH inhibitors are being studied in early-phase trials as well. LY3410738, an oral agent already considered to be more potent and possibly more durable than ivosidenib [29], is currently undergoing phase 1 investigation in specific IDH mutations in BTC (IDH1-R132, IDH2-R140, and IDH2-R172) (NCT04521686). IDH305 is an experimental oral agent exclusively for malignancies harboring IDH1-R132 mutations, with ongoing trials testing its tolerability at this time (NCT02381886). Olutasidenib, an oral inhibitor of mutated IDH1, has demonstrated safety and tolerability as well as modest clinical activity, with 23% of iCCA patients showing stable disease in preliminary data [30]. Trials evaluating its clinical efficacy and updated safety are ongoing. Finally, vorasidenib, an oral dual IDH1/IDH2 inhibitor is also in development. Preliminary studies have shown clinical activity in gliomas, but it has not yet been studied specifically in BTC [31].

### 2.3. Human Epidermal Growth Factor Receptor 2 (HER2) Inhibitors

The overexpression of HER2, also known as ERBB2, is seen in approximately 15–20% of cases of BTC, with a strong predilection for eCCA and GBC [32]. The four HER receptors (HER1-4) dimerize to activate the HER pathway, which depending on the cellular context, produces many outcomes, including growth, differentiation, and cellular migration [33]. Overexpression of one receptor leads to more heterodimerization in general, and to more biased heterodimerization towards more mitogenic and transforming combinations such as HER2 and HER3.

The monoclonal antibody combination of trastuzumab and pertuzumab has been best studied. While it is not FDA-approved, it is incorporated into the NCCN guidelines as an option for patients with HER2-positive tumors after first-line therapy [14]. 

The MyPathway study (NCT02091141) is a large, multi-basket phase 2 clinical trial, one arm of which evaluates the efficacy of pertuzumab and trastuzumab in metastatic biliary tract cancers with HER2 amplification and overexpression that have failed prior therapy. While the initial results were inconclusive [34], with time and increased enrollment, the latest updates present an ORR of 23% (95% CI 11–39) [35]. Additionally, researchers observed that resistance to HER2-targeted therapy may have arisen in the setting of emerging co-alterations, including KRAS and PIK3CA. Future randomized controlled trials hope to solidify this chemotherapy-free regimen for the second line and beyond, as well as to investigate the relationships between genetic aberrations that may confer resistance.

Trastuzumab has also been studied in combination with folinic acid, fluorouracil, and oxaliplatin (FOLFOX) for second line therapy in those who progressed on first–line gemcitabine and cisplatin [36]. Data from a South Korean phase 2 trial showed that of the 34 patients enrolled, 10 had partial response, and 17 had stable disease, resulting in an ORR of 29.4% (95% CI 16.7–46.3) and a disease control rate (DCR) of 79.4% (95% CI 62.9–89.9). Adverse events primarily included neutropenia and anemia, with some grade 3 peripheral sensory neuropathy, but the treatment was overall well–tolerated, with no change in quality of life based on the EuroQoL-VAS score. 

Other anticipated studies include the ongoing phase 2 HERB trial in Japan, which assesses the safety and efficacy of trastuzumab deruxtecan (T-DXd) in patients with HER2–mutant biliary tract cancers who were intolerant or refractory to first-line gemcitabine-containing therapy [37]. T-DXd is an antibody-drug conjugate consisting of the anti-HER2 antibody trastuzumab, a cleavable linker, and a topoisomerase inhibitor, deruxtecan. The preliminary results presented at the 2022 ASCO Annual Meeting showed promise, with an ORR of 36.4% (90% CI 19.6–56.1) and a DCR of 81.8% (95% CI 59.7–94.8) [38]. However, important adverse events were identified, including the development of ILD in 25% of patients and significant gastrointestinal toxicity and myelosuppression, which require further exploration and monitoring.

## 3. Tumor-Agnostic Treatments

### 3.1. BRAF/MEK Inhibitors

BRAF refers to the gene encoding the B-RAF protein, belonging to the RAF (rapidly accelerated fibrosarcoma) family of serine-threonine protein kinases, which serve as the initiating enzymes in the three-tiered Raf/MEK/ERK pathway [39]. Overactivity of this pathway is a well-studied contributor to tumorigenesis. The BRAF V600E mutation, a single point mutation that substitutes a glutamic acid for valine at residue 600, accounts for about 90% of activating BRAF mutations among human tumors [40]. V600E substitution mimics phosphorylation of the activating segment, leading to constitutive ERK signaling.

Activating mutations of BRAF are uncommon in BTC, found in just 5% of cases, primarily in iCCA [41]. When present in BTC, the oncogene confers excellent response rates to combination therapy with the BRAF inhibitor dabrafenib and the MEK inhibitor trametinib [42]. The ROAR trial (NCT02034110) demonstrated the efficacy and safety of this combination in a large phase 2 basket trial of patients with rare solid tumors harboring BRAF V600E mutations, who progressed on prior systemic therapy. Overall response was achieved in 51% of patients (95% CI 36–67) with common adverse events including elevated liver enzymes, and serious treatment-related adverse events occurring in 21% of patients, most often pyrexia [42]. This study led to accelerated FDA approval of dabrafenib plus trametinib as a second line option for BRAF–V600E–mutant biliary cancers as well as for all solid tumors [43].

Phase 1 trials are currently ongoing investigating the safety and preliminary anti-tumor activity of a novel oral small molecule inhibitor of BRAFV600E, ABM-1310, both as monotherapy (NCT05501912) and in combination with MEK inhibitor cobimetinib (NCT04190628). Targeted agents for non-V600E BRAF-mutant tumors are also being studied in melanoma patients, and preliminary results demonstrate promising efficacy of the approved BRAF/MEK inhibitor combination regimens already being used in BRAF V600E-mutant melanoma [44].

### 3.2. Tropomyosin Receptor Kinase (TRK) Inhibitors

Neurotrophic tropomyosin receptor kinase (NTRK) genes, including NTRK1, NTRK2, and NTRK3, encode TRKs, including TRKA, TRKB, and TRKC, which are receptor tyrosine kinases that play a role in the normal development of the nervous system [45]. TRK receptors are typically activated by neurotrophins, which lead to cellular proliferation and growth via ERK signaling. However, fusion rearrangements form hybrid genes between the NTRK sequences juxtaposed to genes that are constitutively activated, leading to ligand-independent signaling of TRK receptors, as well as uncontrolled ERK activation [46].

By this mechanism, TRK fusion proteins are known to be oncogenic drivers of a wide variety of tumors, found significantly more often in rare cancer types, such as secretory breast carcinoma where they have been found in more than 90% of cases [47], compared to <1% in common cancers such as colorectal and lung [48]. In biliary tract cancer, their prevalence has been estimated at around 0.25% [48].

Many TRK inhibitors are being investigated, but entrectinib and larotrectinib are the furthest along in clinical development [46]. Both are approved for NTRK gene fusion-positive tumors both in the first line and after progression on first-line therapy [14]. Larotrectinib was granted accelerated FDA approval [49] after three clinical basket trials including adults and children demonstrated an ORR of 75% (95% CI 61–85) with primarily grade 1 adverse events [50]. Updates on the efficacy and safety of larotrectinib were presented as an abstract at the 2022 ASCO Annual Meeting, confirming an ORR of 69% (95% CI 63–75) with a duration of response (DOR) of 33 months (95% CI 27.3–41.7) [51]. TRAEs were mostly grade 1–2, with 20% experiencing grade 3–4 events, including elevated liver enzymes, neutropenia and lymphopenia. Nevertheless, only 2% discontinued treatment due to adverse effects. 

Similarly, entrectinib was granted accelerated approval by the FDA [52] after the STARTRK-1 and ALKA-372-001 trials demonstrated promising efficacy and safety, with an ORR of 57% (95% CI 43.2–70.8), including 4 complete responses and 27 partial responses [53]. The most common grade 3-4 TRAEs were weight gain and anemia. Recent updates regarding entrectinib’s safety and efficacy were presented at the 2022 ASCO Meeting showing an ORR of 61.3% (95% CI 52.0–70.1) and 25 complete responses, with a median DOR of 20 months (95% CI 13.2–31.1) [54]. The most frequent adverse events were dysgeusia, diarrhea, and weight gain; adverse events resulted in dose interruption in 33% of patients, dose reduction in 24%, and treatment discontinuation in 7.2%.

Ongoing studies seek to evaluate the efficacy and safety of the second-generation agents selitrectinib and repotrectinib, TRK inhibitors that may be able to overcome on-target treatment resistance caused by mutation of the TRK kinase domain itself [55,56,57].

### 3.3. RET Inhibitors

The RET proto-oncogene encodes a receptor tyrosine kinase that is known to activate various downstream cancer effectors [58]. Activating aberrations in RET can result in ligand-independent kinase activation through mutations, fusions/rearrangements, or amplifications. Approximately 1.8% of all cancers possess aberrations in RET, 38% of which are mutations, followed by fusions (31%), and amplifications (25%) [59]. Although RET aberrations are generally rare, many who possess them have had good response to RET-directed therapy. 

Selpercatinib is a highly selective RET kinase inhibitor, approved by the FDA to treat any RET-mutated solid tumors [60]. This came after the results of the LIBRETTO-001 trial (NCT03157128), an ongoing phase ½ open-label basket trial evaluating selpercatinib in RET fusion-positive solid tumors of primary any site [61]. The ORR was 43.9%, with a DOR of 24.5 months (95% CI 9.2-not evaluable), and PFS of 13.2 months (95% CI 7.4–26.2). This tumor-agnostic population included two patients with BTC, one of whom was evaluable and did have a response, while still on treatment at the time of data cutoff. The most common grade 3 TRAEs included hypertension (22%) and elevated liver enzymes (16% ALT, 13% AST). Currently, selpercatinib maintains an NCCN guideline recommendation for the treatment of RET-positive BTC in the setting of progressive disease, and is listed as an option in the first-line setting [14].

Pralsetinib is another potent selective RET inhibitor that earned FDA approval [62] in the setting of RET-positive non-small cell lung cancer and thyroid cancer following the results of the phase 1/2 ARROW study (NCT03037385). The same study went on to assess response to pralsetinib in a diverse group of RET-positive tumors and was also able to demonstrate tumor-agnostic efficacy. The ORR was 57% (95% CI 35–77), including three patients achieving complete responses, and 10 with partial responses [63]. Among this cohort, three patients had BTC; two had confirmed responses, including one who had previously only seen a best response of progressive disease on three prior lines of therapy. Around 86% of patients experienced TRAEs, 69% of which were grade 3 or higher. Common TRAEs included elevations in AST (38%) and ALT (34%), and neutropenia (34%). The FDA has yet to approve pralsetinib in the tumor-agnostic setting; however it has been incorporated into the NCCN guidelines as an option for RET-positive BTC both in the first-line and subsequent-line settings [14].

**Table 1 cancers-15-02105-t001:** Approved targeted therapies and updated trial results.

Target	Targeted Drug	Trial Name(s)	Cohort Size (n)	ORR	mPFS	mOS	% w/TRAE Grade ≥ 3
FGFR2	Pemigatinib [16]	FIGHT-202	146	35.5%	6.9 months	21.1 months	64%
	Futibatinib [18]	FOENIX-CCA2	203	41.7%	8.9 months	20 months	73.1%
	RLY-4008 [21]	ReFocus	38	63.2%	Not reported	Not reported	Not reported
IDH1	Ivosidenib [26,27]	CLARIDHY	187	2.4%	2.7 months	10.3 months	53%
HER2	Trastuzumab/Pertuzumab [34]	MyPathway	39	23%	4 months	10.9 months	46%
	Trastuzumab/Deruxtecan [37]	HERB	22	36.4%	4.4 months	7.1 months	81.3%
BRAF/MEK	Dabrafenib/Trametinib [41]	ROAR	43	51%	9 months	14 months	56%
TRK	Entrectinib [52,53]	STARTRK-1/2 plus ALKA-372-002	150 (17 tumor types)	61.3%	13.8 months	37.1 months	Not reported
	Larotrectinib [49,50]	LOXO-TRK-14001 SCOUT NAVIGATE	206 (25 tumor types)	69%	29.4 months	Not reached	20%
RET	Selpercatinib [60]	LIBRETTO-001	41 (two BTC)	43.9%	13.2 months	18 months	38%
	Pralsetinib [62]	ARROW	23 (three BTC)	57%	7.4 months	13.6 months	72%

### 3.4. Multikinase Inhibitors

Lenvatinib and regorafenib are oral multikinase inhibitors of vascular endothelial growth factor receptor (VEGFR), fibroblast growth factor receptor (FGFR), platelet-derived growth factor receptor (PDGFR), and the RET and KIT signaling networks [64,65]. They work by inhibiting several pathways to inhibit tumor growth and proliferation from multiple angles, with the goal of more comprehensive antitumor activity [66].

Regorafenib is included in the NCCN guidelines as an option for second-line monotherapy in refractory BTC [14]. This recommendation is based on a phase II trial reporting a DCR of 56%, with overall survival rates of 40% at 12 months and 32% at 18 months [67]. Toxicity was seen in 40% overall, with the most common grade 3–4 toxicities being hypophosphatemia in 21%, hypertension in 18%, and hand-foot syndrome in 7%. More recently, the efficacy of regorafenib was further evaluated through the REACHIN trial, which enrolled 66 patients with BTCs of all sites, including iCCA, eCCA, and GBC [68]. The trial demonstrated a stable disease rate of 74% (95% CI 59–90%) but no partial or complete responses. The median PFS was 3.0 months vs 1.5 in the placebo group (95% CI 2.3–4.9 months), with no new safety signals reported, Given this, regorafenib does not have FDA approval for the treatment of BTC, and is used infrequently in clinical practice. It does however hold approval for the treatment of advanced hepatocellular carcinoma (HCC), where a meta-analysis of several trials showed a pooled ORR of 10.1% and DCR of 65.5% in HCC [69].

Trials assessing lenvatinib have also shown modest activity. One phase 2 trial of lenvatinib as second–line monotherapy in BTC included 17 patients, and demonstrated an ORR of 6% and a DCR of 82% [70]. This was further supported by another cohort of 26 patients treated with lenvatinib monotherapy, which demonstrated an ORR of 11.5% with a DCR of 85% [71]. Lenvatinib has also been studied in combination with pembrolizumab. This combination was assessed in a 31-patient arm of the LEAP-005 trial, which demonstrated an ORR of 10% (95% CI 2–26%) [72]. Grade 3 or higher adverse events occurred in 15% of patients, but led to discontinuation of therapy in only 2%. Based on these data, the NCCN guidelines include the combination of lenvatinib and pembrolizumab as a subsequent-line therapy option after progression on first-line therapy [14]. An ongoing clinical trial (NCT04550624) seeks to further validate this combination in a larger cohort. Now that durvalumab has entered the first-line setting, it is unclear where the lenvatinib and pembrolizumab will fit into the treatment landscape.

### 3.5. High Tumor Mutational Burden and Immunotherapy

Programmed cell death ligand 1 (PD-L1) is a ligand expressed in human cells, which activates a physiological immune checkpoint to maintain self-tolerance and prevent auto-immunity [73]. Cancers evolve to express this ligand, allowing for immune evasion. Immunotherapy refers to monoclonal antibody inhibitors of PD-1, the receptor of PD-L1 found on T cells, which block this immunosuppressive checkpoint, allowing host T cells to recognize and destroy cancer cells.

Hypermutated tumors, such as those with high levels of microsatellite instability (MSI-H) or those with a high tumor mutational burden (TMB), defined as ≥10 mutations/Mb, are rich in neoantigens, making them highly immunogenic [74].

Pembrolizumab is a well-studied immune checkpoint inhibitor of PD-1, approved for all MSI-H tumors regardless of tissue origin, marking the first tumor-agnostic FDA approval in history [75]. However, MSI-H positivity is rare, accounting for approximately 1% of all tumors [76], and around 1–2% of all BTCs [77,78]. High TMB may be a more inclusive marker of hypermutation, with MSI-H tumors likely representing a subset within them, although not mutually inclusive [79]. 

Studies have estimated that approximately 25% of tumors have a high TMB [80,81], and pembrolizumab also holds a tissue-agnostic approval for all high–TMB tumors in the case of disease progression [82]. This was based on the results of the KEYNOTE-158 trial (NCT02628067), which looked at 10 cohorts of patients with high TMB tumors of varying sites, and found an ORR of 29% (95% CI 21–39), with 57% reaching a DOR of 12 months or greater, and 50% reaching 24 months or greater [83]. 

This approval was somewhat controversial [84] as many researchers called into question the generalizability of a universal TMB cutoff of 10 mutations/Mb to define high TMB across all tissue sites [80,85]. Further studies attempting to validate TMB as a predictive marker for response to immunotherapy in all cancer types were unable to prove differences in ORR based on TMB alone [86]. In the context of BTC, although the KEYNOTE-158 trial included 63 BTC patients, none had high TMB; nevertheless, objective response was seen in 2 of the 63 patients. Ultimately, oncologists and patients must use good clinical judgment to evaluate the risk-benefit ratio of this approval [87].

### 3.6. Bifunctional Antibody Therapy

Bintrafusp alfa (M7824) is a bifunctional fusion protein composed of a monoclonal antibody targeting PD-L1, fused to an extracellular domain of human transforming growth factor beta (TGF-β) receptors, functioning as a trap for TGF-β within the tumor microenvironment [88]. By blocking this immunosuppressive cytokine, bintrafusp alfa is theorized to prevent progression and metastasis in BTC, and it may represent a novel treatment approach. 

A phase 1 study assessed BTC patients who progressed after first-line chemotherapy, reporting an ORR of 20% (95% CI 8–39), with mOS of 12.7 months (95% CI 6.7–15.7) [89]. All but one of the six responders had a DOR of greater than 12 months. A larger trial ongoing in Germany looked at 159 patients with BTC who had progressed on first-line chemotherapy [90]. The results thus far show an ORR of 10.1% (95% CI 5.9–15.8) and a mOS of 12.7 months (95% CI 6.7–15.7), with the final results still awaited.

## 4. Future Directions

Numerous areas of research show promise for the treatment of BTC. In terms of anti-cancer therapies, agents targeting the PIK3/mTOR/AKT pathway have been studied, due to its importance in oncogenesis across tumor types including BTC [91]. The mTOR inhibitor everolimus has been evaluated in the second line setting for chemo-refractory BTC with modest results [92]. Other large trials of PI3K inhibitors and AKT inhibitors have not reported specific data for BTC patients [93]. Ultimately, agents targeting this pathway have not yet made their way into clinical practice, but studies are ongoing.

Other potentially targetable pathways include those that induce hypoxia, which is known to enable the rapid growth of tumors including BTC [94]. The activation and overexpression of hypoxia-inducible factor 1 (HIF-1) has been associated with increased tumor volume and poor prognosis [95]. Studies seeking to validate this pathway as a therapeutic target are underway, and these agents may make their way into clinical practice in the future.

Similarly, the pathways implicated in autophagy are being studied as potential therapeutic targets in BTC. Autophagy refers to a process by which cancer cells self-degenerate, which can be both carcinogenic and self-destructive [96]. Preclinical studies have demonstrated autophagy’s role in carcinogenesis in cholangiocytes specifically, representing a powerful potential anti-cancer target [97]. Numerous anti-cancer agents are known to therapeutically inhibit or activate autophagy, including those targeting the PIK3/mTOR pathway, as discussed above. However, other agents are theorized to have potenial as autophagy modulators, including autophagy activators such as metformin, loperamide, and amiodarone, as well as autophagy inhibitors such as azithromycin, hydroxychloroquine, and clomipramine [96]. Research is ongoing to better understand the role of autophagy modulation in the treatment of BTC.

Glutamine metabolism is another potentially targetable pathway, with glutamine playing an important role in regulating multiple important signaling pathways and, ultimately, tumor growth [98]. Trials are already underway investigating the utility of Telaglenastat, a glutaminase (GLS1) inhibitor, in the treatment of several solid tumors, including metastatic renal cell carcinoma, colorectal cancer, non-small cell lung cancer, and triple-negative breast cancer [99]. No specific trial is looking into the role of GLS1 inhibitors in BTC at this time. However, studies have found that GLS1 is overexpressed in iCCA, and that downregulation of GLS1 expression may suppress cell invasion and migration [100], making glutamine-directed agents potential future therapies for BTC.

Finally, novel targeted therapies aimed at eradicating cancer stem cells represent important potential methods for treatment-resistant BTC. With multiple signaling pathways involved in the development and survival of cancer stem cells, the agents targeting those pathways themselves, including Notch, Wnt and Hedgehog signaling, have produced only modest effects [101]. The Hippo/yes-associated protein 1 (YAP1) signaling pathway has emerged as a promising target, but no clinical studies have been conducted using Hippo/YAP1 inhibitors to date. Future studies are needed to fully elucidate the potential of this therapeutic strategy.

**Figure 1 cancers-15-02105-f001:**
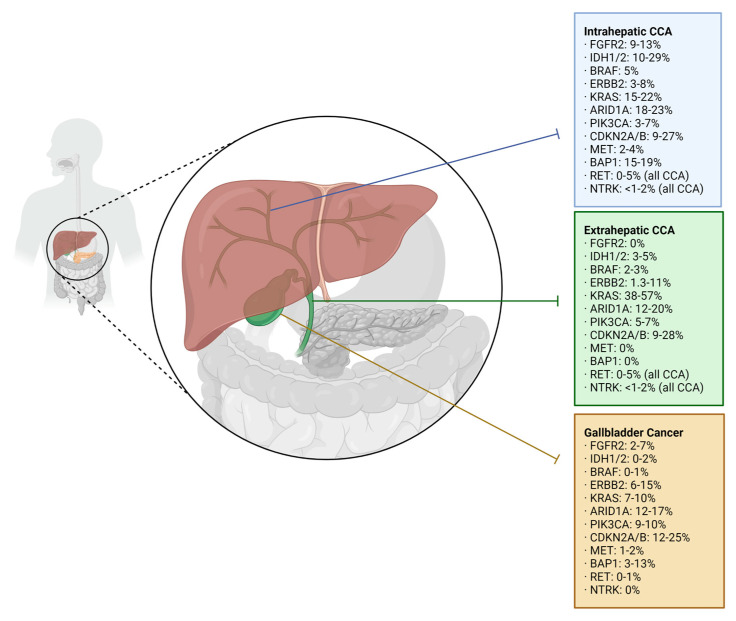
Prevalence of genomic aberrations by anatomic classification in cholangiocarcinoma [91,102,103,104,105,106,107,108,109,110,111].

## 5. Conclusions

Biliary tract cancer remains a deadly disease in need of more treatment options, both within and outside the realm of systemic therapy. As precision medicine continues to change the paradigm of disease management, targeted therapies are emerging as increasingly favorable options in the second-line and beyond. Therefore, molecular profiling plays an important role in determining the ideal treatment course for each individual patient. Likewise, continued awareness of the ever-evolving landscape of therapeutic options is imperative in managing these challenging malignancies.

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
