# Peer review of "Precision Oncology Targets in Biliary Tract Cancer"

_cancers, 2023, doi:10.3390/cancers15072105_

Round 1

Author Response

Reviewer 1:

  1. Should add more details of affiliation for example city and country.
    1. Answer: Thank you for this suggestion. This was added per reviewer’s request. 
  2. In simple summary “Cholangiocarcinoma is a rare but highly lethal cancer” should revise this sentence according to cholangiocarcinoma is not rare in some region particularly in Thailand and Mekong subregions. This is jumped to conclude that.
    1. Answer: Thank you for this comment. This was revised per reviewer’s request. (Line 7)
  3. Regarding from anatomical position in AJCC8, why the author categorized genetic aberrations in tree groups intrahepatic, extrahepatic CCA and gall bladder cancer?
    1. Answer: Thank you for this comment. We have made edits, please see lines 28-30. We chose this classification because the data show that from a genomics perspective, intrahepatic and gallbladder cancers clear represent different entities, but extrahepatic/perihilar bile duct cancer and distal bile duct cancers are extremely similar and can be classified into one group. The surgical differences that apply to early stage disease are not as relevant in the genomics and advanced stage settings.
  4. Should add more data of survival rate in iCCA, eCCA patients.
    1. Answer: This was added, please see lines 36-40.
  5. Table 1. Approved targeted therapies and updated trial results. Should be moved above the table.
    1. Answer: This was addressed per reviewer’s request.
  6. Add the reference in each row of target therapy.
    1. Answer: Thank you for this suggestion, this was added.
  7. From the genetic alterations, the other ARID1A is also moderate-high prevalence in either iCCA and eCCA. Should add more details of this targeted mutation.
    1. Answer: We agree that ARID1A is a relatively highly prevalent mutation in iCCA and eCCA. To reflect this, we have raised it in the list of mutations displayed in the figure. However, since there are no approved therapies targeting ARID1A and we intend this to be a clinically useful review, we have not discussed it further.

Reviewer 2 Report

This review summarizes promising new developments in the treatment of CCA and provides updates on the therapeutic alternatives that are presently available. For individuals with cholangiocarcinoma (CCA) who do not respond to first-line therapies, targeted medicines are becoming an option. These treatments focus on certain CCA oncogenes such FGFR2, IDH, BRAF, and HER-2. CCA also exhibits other genetic abnormalities such RET fusions, TRK fusions, and elevated TMB that are typical with solid tumors.

Comments

a-In Introduction, please also discuss the limitations of the cholangiocarcinoma therapies, which include:

1-Cholangiocarcinoma is often identified after it has progressed, which makes therapy more difficult.

2-Limited treatment options: Surgery, radiation therapy, chemotherapy, and targeted therapy are the only treatments available for cholangiocarcinoma. However, certain individuals may not be eligible for particular therapies because of their health state, and these treatments may not always be successful.

4-Recurrence: Cholangiocarcinoma has a significant probability of returning following therapy, which makes management difficult.

b- In addition to the HER2, IDH1 and FGFR, the authors should also discuss about the VEGF: A cytokine involved in angiogenesis, vascular endothelial growth factor (VEGF) is overexpressed in several malignancies, including CCA. Anti-VEGF medicines are being looked at as a possible course of action.

c- There are several metabolic targets in CCA treatments, which is also need to be discussed in the text, including.

1-Glutamine metabolism: The amino acid glutamine is increasingly important to CCA cells as a source of energy. Drugs that specifically target the glutamine metabolism-related enzymes have been developed as a result of this.

2-Autophagy is a cellular mechanism that helps keep cellular energy levels stable by destroying damaged cellular components. Autophagy is a possible target for treatment since it has been shown in CCA to contribute to tumor survival and resistance to chemotherapy.

3-The PI3K/AKT/mTOR pathway, which promotes cellular survival and expansion, is often active in CCA. In preclinical investigations, medications that target this pathway have showed promise in the treatment of CCA.

4-Hyperoxia-Inducible Factor 1 (HIF-1) is a transcription factor that controls the expression of genes related to angiogenesis and glucose metabolism. In preclinical investigations of CCA, HIF-1 inhibition has shown effectiveness.

5-Cancer stem cells: It has been suggested that cancer stem cells are responsible for the development and medication resistance of CCA. Although more study is required to properly understand their function in the illness, targeting these cells is a viable approach for the therapy of CCA.

Author Response

Reviewer 2

  1. In Introduction, please also discuss the limitations of the cholangiocarcinoma therapies, which include:
    1. Cholangiocarcinoma is often identified after it has progressed, which makes therapy more difficult.
      1. Answer: This was added (line 41)
    2. Limited treatment options: Surgery, radiation therapy, chemotherapy, and targeted therapy are the only treatments available for cholangiocarcinoma. However, certain individuals may not be eligible for particular therapies because of their health state, and these treatments may not always be successful.
      1. Answer: This was added (lines 43-48)
    3. Recurrence: Cholangiocarcinoma has a significant probability of returning following therapy, which makes management difficult.
      1. Answer: Comment was added (lines 54-56 and 65-68)
  2. In addition to the HER2, IDH1 and FGFR, the authors should also discuss about the VEGF: A cytokine involved in angiogenesis, vascular endothelial growth factor (VEGF) is overexpressed in several malignancies, including CCA. Anti-VEGF medicines are being looked at as a possible course of action.
    1. Answer: Thank you very much for this comment. Information was added in lines 306-336
  3. There are several metabolic targets in CCA treatments, which is also need to be discussed in the text, including:
    1. Glutamine metabolism: The amino acid glutamine is increasingly important to CCA cells as a source of energy. Drugs that specifically target the glutamine metabolism-related enzymes have been developed as a result of this.
      1. Answer: Thank you very much for this comment. We have included a section on future directions of therapy including this topic (lines 410-419)
    2. Autophagy is a cellular mechanism that helps keep cellular energy levels stable by destroying damaged cellular components. Autophagy is a possible target for treatment since it has been shown in CCA to contribute to tumor survival and resistance to chemotherapy.
      1. Answer: Thank you very much for this comment. We have included a section on future directions of therapy including this topic (lines 397-408)
    3. The PI3K/AKT/mTOR pathway, which promotes cellular survival and expansion, is often active in CCA. In preclinical investigations, medications that target this pathway have showed promise in the treatment of CCA.
      1. Answer: Thank you very much for this comment. We have added relevant information in lines 384-390.
    4. Hyperoxia-Inducible Factor 1 (HIF-1) is a transcription factor that controls the expression of genes related to angiogenesis and glucose metabolism. In preclinical investigations of CCA, HIF-1 inhibition has shown effectiveness.
      1. Answer: Thank you very much for this comment. We have included a section on future directions of therapy, including this topic (lines 391-396)
    5. Cancer stem cells: It has been suggested that cancer stem cells are responsible for the development and medication resistance of CCA. Although more study is required to properly understand their function in the illness, targeting these cells is a viable approach for the therapy of CCA.
      1. Answer: Thank you very much for this comment. We have included a section on future directions of therapy, including this topic (lines 420-427).

Reviewer 3 Report

This manuscript has summarized the current progress of targeted therapies in the treatment of CCA. Here are some points of concern:

The authors have included gallbladder carcinoma as one type of cholangiocarcinoma in the introduction. I’m not sure if the authors have confused cholangiocarcinoma with biliary tract cancers. Gallbladder cancer as a type of biliary tract cancer is often included in the research of cholangiocarcinoma, but it is usually treated as a separate disease from cholangiocarcinoma. I think clarification on this point is necessary. In addition, according to reference 1, it is perihilar CCA instead of GBC that is accounting for approximately 50–60% of all CCAs.

This manuscript has summarized the clinical test results of different drugs for targeted therapy. I appreciate the work of information collection but I think it would be helpful to make some comparison and discuss more about the pros and cons of different inhibitors in clinical application.

This manuscript failed to discuss about the future direction of the research of precision medicine in CCA treatment.

Author Response

Reviewer 3:

  1. The authors have included gallbladder carcinoma as one type of cholangiocarcinoma in the introduction. I’m not sure if the authors have confused Cholangiocarcinoma with biliary tract cancers. Gallbladder cancer as a type of biliary tract cancer is often included in the research of cholangiocarcinoma, but it is usually treated as a separate disease from cholangiocarcinoma. I think clarification on this point is necessary. In addition, according to reference 1, it is perihilar CCA instead of GBC that is accounting for approximately 50–60% of all CCAs.
    1. Answer: We agree that this is a confusing issue in the literature, but we want to be inclusive in a review. In advanced stages (which a review on genomic targets and treatments will be most applicable to), the distinction between cholangiocarcinoma and biliary tract cancers is often blurred in prior literature and not as useful from a therapeutic perspective, for which reason we have included gallbladder cancers.
  2. This manuscript has summarized the clinical test results of different drugs for targeted therapy. I appreciate the work of information collection but I think it would be helpful to make some comparison and discuss more about the pros and cons of different inhibitors in clinical application.
    1. Answer: Thank you very much for this comment. Head-to-head comparisons of these agents are not yet available, therefore we cannot make conclusions or recommendations between them. As a literature review article, we want to be careful about making treatment comparisons that could be misconstrued as treatment recommendations or medical advice.
  3. This manuscript failed to discuss about the future direction of the research of precision medicine in CCA treatment. (This is not within the scope of the paper, we mostly focus on the available modalities).
    1. Answer: We have added a section regarding future direction (please see lines 384-427)

Reviewer 4 Report

Very interesting and comprehensive review on the state of art of precision oncology in cholangiocarcinoma. I would suggest to add a quick comment on the role of regorafenib in the therapy of this tumor, with also a reference to its use in liver malignancy (cite the recent meta-analysis PMID: 31877664)

Maybe some more tables could be useful for the reader. 

Author Response

Reviewer 4:

  1. Very interesting and comprehensive review on the state of art of precision oncology in cholangiocarcinoma. I would suggest to add a quick comment on the role of regorafenib in the therapy of this tumor, with also a reference to its use in liver malignancy (cite the recent meta-analysis PMID: 31877664)
    1. Answer: Thank you for your suggestion. Information about regorafenib was added in lines 307-325. The suggested citation was included.
  2. Maybe some more tables could be useful for the reader.
    1. Answer: Thank you very much for this comment. We have included a comprehensive table with all available trials and outcomes as well as a figure with the prevalence of genomic mutations in CCA.

Round 2

Reviewer 2 Report

All the comments have now been addressed by the Authors.

Author Response

Thank you for your thoughtful edits

Reviewer 3 Report

If the authors want to be inclusive for this review as they have explained, why not just use the term “biliary tract cancer” instead of “cholangiocarcinoma” for this review since the prior is free from confusion? Also, I don’t see any of the clinical trials discussed in this paper had included gallbladder cancer as a type of cholangiocarcinoma. They either focused on cholangiocarcinoma or referred to biliary tract cancers when gallbladder cancer was included.

Author Response

  1. If the authors want to be inclusive for this review as they have explained, why not just use the term “biliary tract cancer” instead of “cholangiocarcinoma” for this review since the prior is free from confusion? Also, I don’t see any of the clinical trials discussed in this paper had included gallbladder cancer as a type of cholangiocarcinoma. They either focused on cholangiocarcinoma or referred to biliary tract cancers when gallbladder cancer was included.
    1. answer: Thank you for your thoughtful commentary. We agree with your reasoning and have chosen to adjust the paper to reflect the inclusive nature of the trials and treatment guidelines. For this reason, unless specifically referring to Cholangiocarcinoma, the terminology has been updated to "biliary tract cancer" or "BTC."